# Sex-specific single transcript level atlas of vasopressin and its receptor (AVPR1a) in the mouse brain

**Anisa Azatovna Gumerova, Georgii Pevnev, Funda Korkmaz, Uliana Cheliadinova, Guzel Burganova, Darya Vasilyeva, Liam Cullen, Orly Barak, Farhath Sultana, Weibin Zhou, Steven Lee Sims, Emily Weiss, Victoria Laurencin, Tal Frolinger, Se-Min Kim, Ki A Goosens, Tony Yuen, Mone Zaidi*, Vitaly Ryu***

Institute for Translational Medicine and Pharmacology, Icahn School of Medicine at Mount Sinai, New York, United States

**\*For correspondence:**
mone.zaidi@mountsinai.org
(MZ);
vitaly.ryu@mssm.edu (VR)

## eLife Assessment

This work presents a brain-wide atlas of vasopressin (Avp) and vasopressin receptor 1A (Avpr1a) mRNA expression in mouse brains using high-resolution RNAscope in situ hybridization. The single-transcript approach provides precise localization and identifies additional brain regions expressing Avpr1a, creating a **valuable** resource for the field. The revised manuscript is clearer and more impactful, with improved figures, stronger data organization, and enhanced scholarship through added context and citations. Overall, the evidence is **compelling**, and the atlas should be broadly of use to researchers studying vasopressin signaling and related neural circuits.

**Abstract** Vasopressin (AVP), a nonapeptide synthesized predominantly by magnocellular hypothalamic neurons, is conveyed to the posterior pituitary via the pituitary stalk, where AVP is secreted into the circulation. Known to regulate blood pressure and water homeostasis, it also modulates diverse social behaviors, such as pair bonding, social recognition, and cognition in mammals, including humans. Importantly, AVP modulates social behaviors in a sex-specific manner, perhaps due to sex differences in the distribution in the brain of AVP and its main receptor AVPR1a. There is a *corpus* of integrative studies for the expression of AVP and AVPR1a in various brain regions, and their functions in modulating central and peripheral actions. In order to purposefully address sexually dimorphic and novel roles of AVP on central and peripheral functions through its AVPR1a, we utilized RNAscope to map *Avp* and *Avpr1a* single transcript expression in the mouse brain. As the most comprehensive atlas of AVP and AVPR1a in the mouse brain, this compendium highlights the importance of newly identified AVP/AVPR1a neuronal nodes that may stimulate further functional studies.

## Introduction

Vasopressin (AVP), a nonapeptide synthesized primarily by magnocellular neurons of the paraventricular nucleus (PVH) and supraoptic nucleus (SO) of the hypothalamus, is conveyed along axons of magnocellular neurons via the pituitary stalk to the posterior pituitary, where AVP is released into circulation to exert hormonal functions (*Brownstein et al., 1980*; *Young and Gainer, 2003*). Involved in the regulation of blood pressure and water balance (*Silva et al., 1969*; *Stockand et al., 2022*), AVP also regulates diverse social behaviors, such as pair bonding, social recognition, and cognition in all mammals, including humans (*Lukas and Neumann, 2013*; *Lukas et al., 2013*; *Veenema and*

*Neumann, 2008*; *Meyer-Lindenberg et al., 2011*; *Albers, 2015*). It should be noted that similar to OXT, AVP is evolutionarily conserved across invertebrates and vertebrate taxa, differing from OXT by just two amino acids. Importantly, AVP modulates social behaviors in a sex-specific manner, perhaps due to sex differences in AVP and its receptor expression in the brain (*Winslow et al., 1993*; *Insel and Hulihan, 1995*; *Dantzer et al., 1987*; *Rigney et al., 2023*; *Landgraf et al., 2003*; *Veenema et al., 2012*).

AVP receptors are G protein-coupled receptors consisting of two major subtypes, the V1 receptor (AVPR1) and V2 receptor (AVPR2) (*Dumais and Veenema, 2016*). AVPR1 has two subtypes, namely AVPR1a and AVPR1b, that mediate the effects of AVP on social behaviors. Avpr1a will be the overarching focus of this study given its abundance and ubiquity in brain regions (*Albers, 2015*). In fact, differences in AVPR1a genetic variability and expression patterns determine specific social phenotypes (*French et al., 2016*; *Phelps et al., 2017*; *Lim et al., 2004*; *Fink et al., 2007*; *Ren et al., 2014*). There is evidence that AVPR1a is involved in maternal care, pair bonding, behavioral aggression, anxiety, social recognition and social play (*Lukas et al., 2013*; *Albers, 2015*; *Lim et al., 2004*; *Bielsky et al., 2004*; *Neumann and Landgraf, 2012*; *Lago et al., 2021*; *Thompson et al., 2006*). The sex-dependent distribution of Avpr1a across the brain in different species provides a proxy for the distribution of AVP binding and, therefore, provides further evidence for central AVP neural nodes of physiological relevance.

Blocking AVPR1a inhibits social recognition in the rat, while AVPR1a knockout mice fail to display social recognition (*Veenema et al., 2012*; *Bielsky et al., 2004*; *Bielsky et al., 2005*). These effects of AVP in enhancing social recognition are mediated via AVPR1a in the lateral septum (*Veenema et al., 2012*; *Bielsky et al., 2005*). Sex differences in Avpr1a binding densities have been described in several brain sites of Wistar rats. Namely, males display higher AVPR1a binding densities in the following forebrain areas: somatosensory and piriform cortex, medial posterior bed nucleus of the stria terminalis (BNST), nucleus of the lateral olfactory tract (LOT), anteroventral thalamic nucleus (VA), tuberal lateral hypothalamus (LH), stigmoid hypothalamus (Stg), and dentate gyrus (DG) (*Dumais and Veenema, 2016*). In male prairie voles, central AVP infusion facilitates selective aggression associated with pair bond formation and partner preference, and the Avpr1a antagonist 1-(β-mercapto-β, β-cyclopentamethylene propionic acid) does not seem to inhibit aggression (*Winslow et al., 1993*). In contrast, central AVP infusion induces partner preference in female prairie voles (*Insel and Hulihan, 1995*). Comparative analysis of AVPR1a distribution has revealed higher densities of AVPR1a binding in the ventral pallidum, amygdala, and thalamus of prairie voles than that of meadow or montane voles (*Insel et al., 1994*; *Wang et al., 1997*). It has also been reported that AVPR1a antagonism specifically in the ventral pallidum prevents mating-induced partner preference in male prairie voles (*Lim and Young, 2004*).

Although osmotically stimulated AVP release with short latency and duration from hypothalamic PVH and SO magnocellular axon terminals occurs in the pituitary (*Summy-Long and Kadekaro, 2001*; *Stricker et al., 2002*), it is also released locally from somata and dendrites in the SO with a longer delay responding to osmotic challenge (*Ludwig et al., 1994*; *Ludwig and Leng, 1997*). Such local release is likely to facilitate autocrine and/or paracrine regulation of SO magnocellular neuronal activity and inhibit further systemic AVP output (*Ludwig and Leng, 1997*; *Wang et al., 1982*). Of note, somatodendritic AVP release in response to direct hypertonic stimulation is attenuated by V1/V2 receptor antagonism, implying that AVP may facilitate its own release by acting on autoreceptors within magnocellular neurons of the SO (*Wotjak et al., 1994*). The importance of autofacilitation to local AVP release may lie in fine-tuned regulation of AVP actions toward physiological demands. Alternatively, AVP could putatively diffuse over longer distances to bind to adjacent receptors. The relative contribution of AVP autoreceptor subtypes, including AVPR1a, to this phenomenon awaits further clarification.

Current studies implicate posterior pituitary hormones, traditionally thought of as master regulators of a single physiological target, in the control of multiple bodily systems, either directly or via their receptors in the brain (*Neumann and Landgraf, 2012*; *Zaidi et al., 2018*; *Carter, 2014*; *Koshimizu et al., 2012*; *Jurek and Neumann, 2018*). Non-traditional actions of AVP include its ability to affect skeletal homeostasis, wherein it negatively regulates osteoblasts and stimulates osteoclasts. This explains the bone loss that accompanies hyponatremia in patients with elevated AVP levels (*Tamma et al., 2013*). Furthermore, we have shown that AVP (via AVPR1a) and oxytocin (via OXTR) have

opposing skeletal actions—effects that may relate to the pathogenesis of bone loss in chronic hypo-natremia, and pregnancy and lactation, respectively (*Tamma et al., 2013*; *Sun et al., 2019*; *Sun et al., 2016*; *Tamma et al., 2009*).

Detecting specific AVPR1a and AVPR1b in the brain has had limitations for a long time due to the availability of only nonselective radioligands, such as tritium labeled ($^3$H) AVP ligands, which bind to both receptors (*Phillips et al., 1990*). Although there is a large body of integrative studies for the expression of AVP and AVPR1a in various brain regions, and their functions in regulating central and peripheral actions, there remains the need for detailed, sex-specific mapping of the AVP/AVPR1a neuronal nodes in the brain. We utilized RNAscope—a technology that detects single RNA transcripts—to create a comprehensive atlas of AVP and AVPR1a in the mouse brain. It may seem somewhat remarkable that newly discovered brain areas for receptors of such evolutionarily conserved and well-studied hormones AVP and OXT are currently emerging, with inferences to novel functions. We believe that this atlas of AVP and its AVPR1a in concrete brain sites at a single transcript level should provide a resource to neuroscientists to deepen our understanding of classical and novel central and peripheral functions of AVP by interrogating AVPR1a site-specifically.

## Results

AVP receptors are G protein-coupled receptors consisting of two major subtypes, AVPR1 and AVPR2 (*Dumais and Veenema, 2016*). AVPR1 is further divided into AVPR1a and AVPR1b receptor subtypes that mediate the effects of AVP in the brain on social behaviors. In this study, we have provided not only a distribution mapping of AVP and AVPR1a in the brain, but also assessed sex differences in their expression by RNAscope. Allowing the detection of single transcripts, RNAscope uses ~20 pairs of transcript-specific double *Z*-probes to hybridize 10-μm-thick whole brain sections. Preamplifiers first hybridize to the ~28 bp binding site formed by each double *Z*-probe; amplifiers then bind to the multiple binding sites on each preamplifier; and finally, labeled probes containing a chromogenic enzyme bind to multiple sites of each amplifier.

RNAscope data were quantified on every tenth section of the whole brains from coded three female and three male mice. For simplicity and clarity in the graphs, a scatter plot has been shown for three nuclei, subnuclei, and regions with the highest AVP and AVPR1a transcript densities, as well as their absolute transcript counts. Each section was viewed and analyzed using CaseViewer 2.4 (3DHIS-TECH, Budapest, Hungary) and QuPath v.0.2.3 (University of Edinburgh, UK). The *Atlas for the Mouse Brain in Stereotaxic Coordinates* (*Paxinos and Franklin, 2007*) was used to identify every nucleus, subnucleus, or region, which was followed by manual counting of *Avp* and *Avpr1a* transcripts by two independent observers (VR and AG) in every tenth section using the tag feature. Receptor density was calculated by dividing the absolute receptor number by the total area (μm$^2$, ImageJ) of every nucleus, subnucleus, or region. The highest *Avp* and *Avpr1a* values in the brain regions are presented as means ± SE and compared with those of the opposite sex. Photomicrographs were prepared using Photoshop CS5 (Adobe Systems) only to adjust brightness, contrast, and sharpness, to remove artifacts (i.e. obscuring bubbles) and to make composite plates.

RNAscope revealed *Avp* expression in the hypothalamus, forebrain, hippocampus, and cortex of both female and male mice (*Figure 1A*); however, *Avp* expression in the 3rd ventricular region and thalamus was found only in the female mouse (*Figure 1B*). Whereas the numbers of *Avp*-expressing cells were greater in females compared with males in the hypothalamus (607 vs 471), 3rd ventricular region (7 vs 0) and thalamus (2 vs 0), those numbers were lower in the hippocampus (58 vs 118), forebrain (50 vs 77) and cortex (5 vs 6) (see Appendix for Glossary and *Figure 1—figure supplement 1* for raw count graphs).

The highest *Avp* transcript densities were detected in the following brain nuclei, subnuclei, and regions of female and male mice, respectively: ventricular region—3V only for females, hypothalamus—SChVL and PaDC, forebrain—MPOM and VLPO, hippocampus—Py and GrDG, thalamus—ic only for females, and cerebral cortex—Pir for both (*Figure 1B*). Representative micrographs of some of the hypothalamic and forebrain regions with highest *Avp* expression are shown in *Figure 1C*.

We found that *Avpr1a* transcript expression in several brain nuclei, e.g., the MS of the forebrain, medullary 12N, IRt, LRt, MdV (*Figure 2—figure supplement 1B*), displayed individual patterns of expression compared with more ubiquitous and even expression noted in most of the other brain areas, suggesting brain-site-specific functional diversity and context-selective regulation of

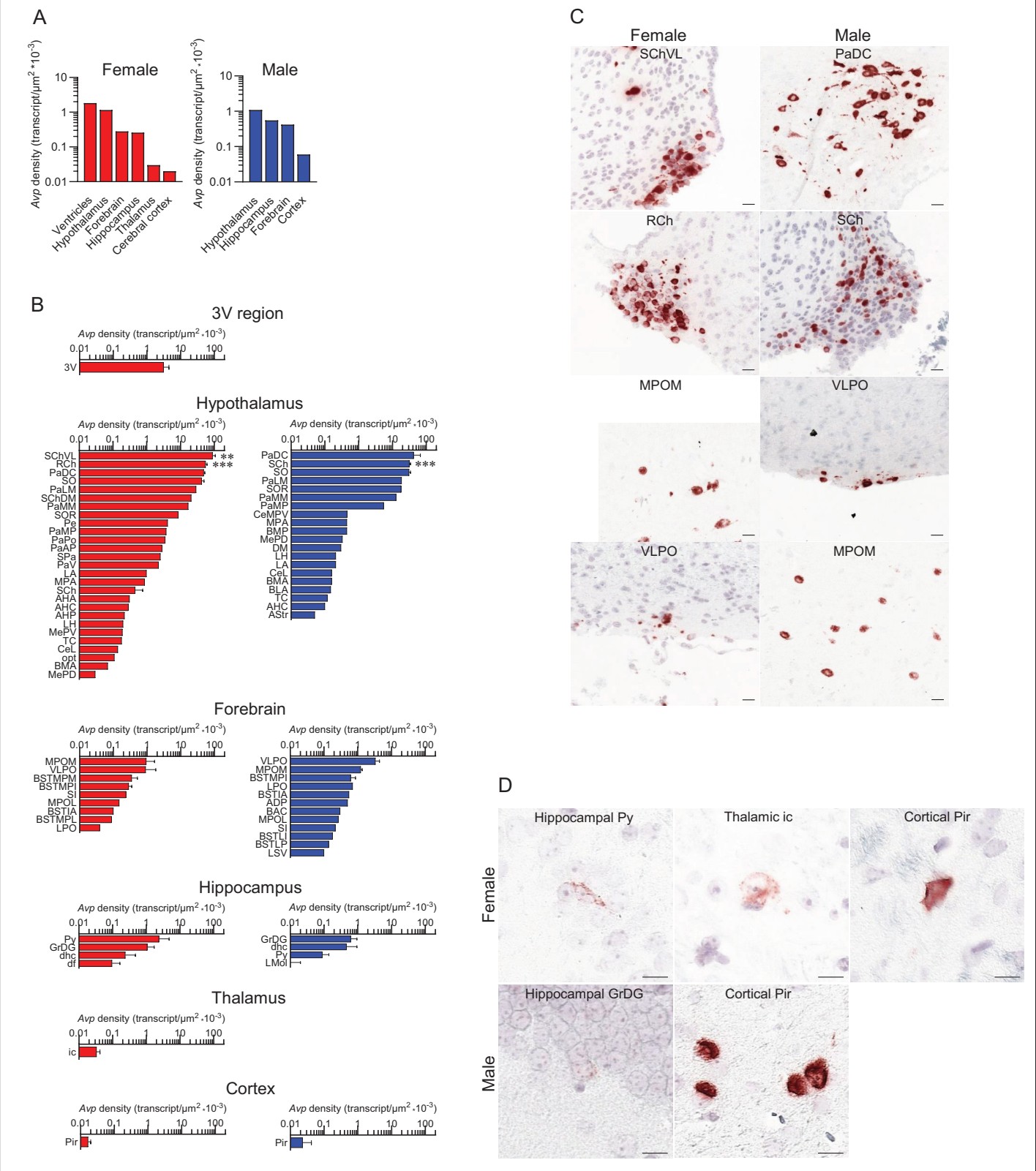

**Figure 1.** Sex-specific *Avp* transcript density in the brain. (**A**) *Avp* transcript density in main brain regions detected by RNAscope. Female 3V region and hypothalamus and male hypothalamus had the highest *Avp* transcript density. (**B**) *Avp* transcript density in nuclei, subnuclei, and regions of the hypothalamus and forebrain. (**C**) Representative micrographs of some of the hypothalamic and forebrain regions with highest *Avp* expression. Scale bar: 20 μm. (**D**) Novel *Avp* transcripts found in nuclei, subnuclei, and regions of the hippocampus, thalamus, and cerebral cortex. Scale bar: 10 μm. Note that

*Figure 1 continued on next page*

*Figure 1 continued*

*Avp* transcripts were found only in the female 3V ependymal layer and thalamus. N=3 mice *per* sex, values are shown as means ± SE . ***p<0.001 and **p<0.01, unpaired Student's *t* test and two-way ANOVA.

The online version of this article includes the following figure supplement(s) for figure 1:

**Figure supplement 1.** Sex-specific *Avp*-positive cells in the brain.

*Avpr1a*-mediated signaling. We report the expression of the *Avpr1a* in 398 and 375 brain nuclei, subnuclei, and regions of the female and male mice, respectively. *Avpr1a* transcripts were detected bilaterally, with no apparent ipsilateral dominance. Probe specificity was established by detecting a positive signal in renal tubules (positive control) with an absent signal in the FrA of the frontal cortex (negative control) (*Figure 2A*). Representative micrographs of the sex-specific medullary CC and hypothalamic Arc with highest *Avpr1a* expression are shown in *Figure 2B*.

In the female, total *Avpr1a* transcript numbers were the highest in the medulla followed in descending order by the hypothalamus, cortex, midbrain and pons, forebrain, thalamus, cerebellum, hippocampus, olfactory bulb, and ventricular regions (*Figure 2—figure supplement 1A*). In the male, *Avpr1a* transcripts were the highest in the forebrain, followed by the medulla, midbrain and pons, hypothalamus, cortex, hippocampus, thalamus, olfactory bulb, cerebellum, and ventricular regions (*Figure 2—figure supplement 1A*). Detailed *Avpr1a* transcript counts are shown in *Figures 3 and 4*.

Using the RNAscope dataset, we further calculated *Avpr1a* density in all brain divisions (*Figure 2C*), nuclei, subnuclei, and regions (*Figures 3 and 4*). Highest *Avpr1a* densities in the female and male mice, respectively, were noted in nuclei, subnuclei, and regions as follows: ventricular regions—CC ependymal region for both with 2.92-fold greater in the male; hypothalamus—Arc for both; medulla—IOBe and InM; midbrain and pons—vtgx and DRI; forebrain—AC and LSI; olfactory bulb—Mi and Tu; hippocampus—TS and df; thalamus—PV and Xi; cortex—MPtA and AID; and cerebellum—1Cb for both with 2.58-fold greater in the female (*Figures 3 and 4*).

In addition, RNAscope analysis revealed that various brain nuclei, subnuclei, and regions in both female and male mice co-localized *Avp* and *Avpr1a* transcripts. *Avpr1a* to *Avp* ratios within the same brain nucleus, subnucleus, and region in both sexes are demonstrated in *Figure 5*. Finally, RNAscope showed *Avp* and *Avpr1a* expression in the posterior pituitary lobe with *Avp* and *Avpr1a* transcript densities that were higher in male compared with female mice, however, without statistical significance (*Figure 6*).

## Discussion

Here, we attempted to integrate previous information on sex-specific AVP and its AVPR1a expression in the murine brain. AVPR1a is the most abundant and widespread receptor in the brain (*Albers, 2015*) that plays a dominant role in regulating behavior. In addition, we focused on paradigm-shifting non-traditional roles of central AVP signaling in light of newly discovered AVPR1a. We report AVP expression in 41 female and 13 male brain nuclei, subnuclei, and regions. Moreover, we identified abundant AVPR1a expression in 398 female and 375 male brain nuclei, subnuclei, and regions. Therefore, this report is the most exhaustive atlas of brain *Avp* and *Avpr1a* expression at the single transcript level.

It has been reported that AVP synthesis and AVP fiber projections are sexually dimorphic in specific brain sites (for review, see: *Dumais and Veenema, 2016*). To our knowledge, the first discovery of the sexually dimorphic nature of AVP in the rat brain was made by *de Vries et al., 1981*. That is, males displayed more AVP-immunoreactive fibers in the lateral septum and lateral habenular nucleus over females (*de Vries et al., 1981*). Surprisingly, sex differences in AVP fiber density in the LS and medial amygdala (MeA) originate from the BNST, given only lesions to the BNST, but not the PVH, result in decreased AVP fiber density in the LS (*de Vries and Buijs, 1983*; *Caffé et al., 1989*). In adult rats, AVP fiber density from the BNST and MeA is dependent on circulating gonadal hormones, as gonadectomy eliminates AVP expression and hormone replacement restores AVP fiber network (*de Vries et al., 1984*; *Miller et al., 1992*). Nonetheless, gonadal hormones appear to only partially explain sex differences in AVP expression because both females and males, exposed to a similar gonadal steroid hormone regime, still differ sexually (*de Vries and al-Shamma, 1990*; *de Vries et al., 1994*).

Magnocellular neurons of the PVN, SO, and SCh of the hypothalamus are the predominant source of AVP synthesis. Hypothalamic AVP synthesis in most rodent species is similar between males and

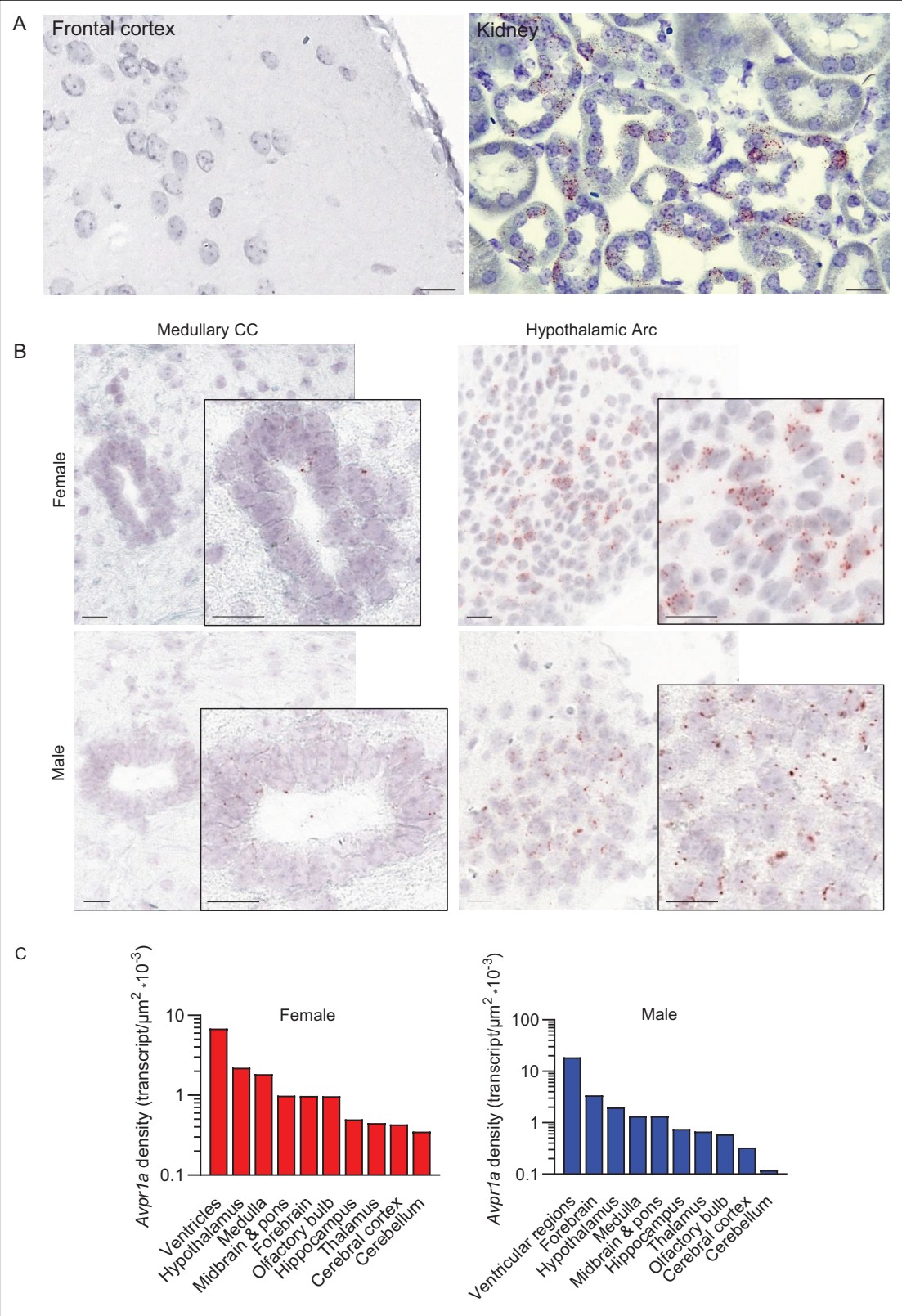

**Figure 2.** Sex-specific *Avpr1a* transcript density in main brain divisions. (**A**) Probe specificity was established by a positive signal in renal tubules of the kidney (positive control) with an absent signal in the frontal association cortex (FrA) (negative control). Sale bar: 20 µm. (**B**) Representative micrographs of medullary central canal (CC) and hypothalamic arcuate nucleus (Arc). (**C**) *Avpr1a* transcript density in main brain regions detected by RNAscope. N=3 mice *per* sex. Scale bar: 25 µm (controls).

*Figure 2 continued on next page*

*Figure 2 continued*

The online version of this article includes the following figure supplement(s) for figure 2:

**Figure supplement 1.** Sex-specific *Avpr1a* transcript numbers in main brain divisions.

females in the PVH and SO of mice (*Joca et al., 2014*; *Steinman et al., 2015*); PVH, SO, and SCh of voles (*Wang, 1995*; *Wang et al., 1996*); PVH, MPOA, LH, and AHA of Mongolian gerbils and Chinese striped hamsters (*Wang et al., 2013*) (for review, see *Dumais and Veenema, 2016*). In concordance with these reports, we also found no sex differences in *Avp* synthesis, as evidenced by similar numbers of *Avp*-expressing neurons in the PVH and SO of mice; however, we did note sex differences in *Avp* expression density in specific hypothalamic nuclei. Notably, *Avp* expression density was higher in several PVH (PaLM and PaMM) and suprachiasmatic (SChVL and RCh) subnuclei of female compared to male mice. No sex differences in SO-*Avp* expression density were found.

Furthermore, *Avpr1a* transcript density was highest in the arcuate nucleus (Arc) and retrochiasmatic subnucleus (RCh) in female compared with Arc and suprachiasmatic nucleus (SCh) of male mice. *Avpr1a* expression in the Arc has previously been reported (*Ostrowski et al., 1992*); however, its role was unclear until a recent report demonstrating a critical involvement of Arc-NPY in the regulation of fluid homeostasis and the induction of salt water-induced hypertension through AVP modulation in the SO (*Zhang et al., 2022*); the latter receives direct projections from the Arc (*Pineda et al., 2016*). In contrast, it is plausible, but by no means proven, that PVH- and/or SO-AVP may modulate Arc anorexigenic neurons to inhibit ingestive behavior, which has previously been shown with PVH oxytocinergic neurons (*Maejima et al., 2014*). Indeed, increasing evidence suggests that AVP reduces feeding in mammals (*Meyer et al., 1989*; *Langhans et al., 1991*).

It has been reported that in the rat, AVP is an important output of the SCh targeting AVP cells in other hypothalamic areas—its release into the CSF peaks in the early morning and declines later in a day (*Kalsbeek et al., 2010*). Specifically, SCh-AVP secreted during late sleep activates osmosensory afferents to AVP neurons in the SO and organum vasculosum of the lamina terminalis (*Trudel and Bourque, 2010*; *Gizowski et al., 2016*). Similar to rodents, studies in humans also determined that the main AVP projections from the SCh target the anteroventral hypothalamic area, sub-PVH, as well as ventral parts of the PVH and DMH—a remarkable evolutionary conservation of SCh innervation from rodent to human (*Dai et al., 1998a*; *Dai et al., 1998b*). The fact that the SCh is another brain nucleus with high AVP and AVPR1a expression density (greater in males vs females) accentuates an important role of SCh-AVP in circadian rhythmicity, notably impacting neuroendocrine day/night rhythms, feeding timing, period, precision, and synchronization of SCh neurons (*Kalsbeek et al., 2010*; *Rohr et al., 2021*; *Yoshimura et al., 2021*).

In the hindbrain, the highest *Avpr1a* transcript density was noted in the inferior olive, beta subnucleus (IOBe) of female mice, and intermedius nucleus (InM) of male mice. It has been reported that AVP fibers are apparent in the hindbrain, such as the parabrachial nucleus, locus coeruleus, and near inferior olive nuclei (*Young et al., 1999*). In this regard, *Avpr1a* mRNA expression has been noted in the inferior olive (*Ostrowski et al., 1992*). Given this nucleus has been implicated in various functions, including learning and timing of movements, it is possible that AVPR1a in the inferior olive may be activated by the paracrine release of AVP from distant nuclei, such as the SCh, to control motor learning and timing. Alternatively, AVPR1a in the inferior olive may respond to other ligands (e.g., OXT) found in nearby regions (*Szczepanska-Sadowska et al., 2021*). The role of AVPR1 in the InM of male mice is less clear, but because the InM sends monosynaptic projections to the NTS (*Edwards et al., 2009*) that is essential for blood pressure control by AVP and receiving information from the cardiovascular receptors (*Zanutto et al., 2010*), a possible coordinated control by hindbrain AVP of blood pressure and cardiovascular function.

Although the midbrain, pons, and forebrain displayed less abundant *Avpr1a* transcript density, they revealed further sex differences. In the midbrain and pons, the highest *Avpr1a* density was observed in the ventral tegmental decussation (vtgx) in females and dorsal raphe nucleus, interfascicular part (DRI), in males. AVP and OXT in the ventral tegmental area are known to regulate social interactions with rewarding properties. Indeed, humans, as inherently social beings, show a strong inclination to affiliate and share their emotions with each other (*Baumeister and Leary, 1995*; *Wagner et al., 2015*). Sex differences in ventral tegmental AVPR1a make biological sense, as social interaction of females, specifically, with pups and, generally, with counterparts throughout

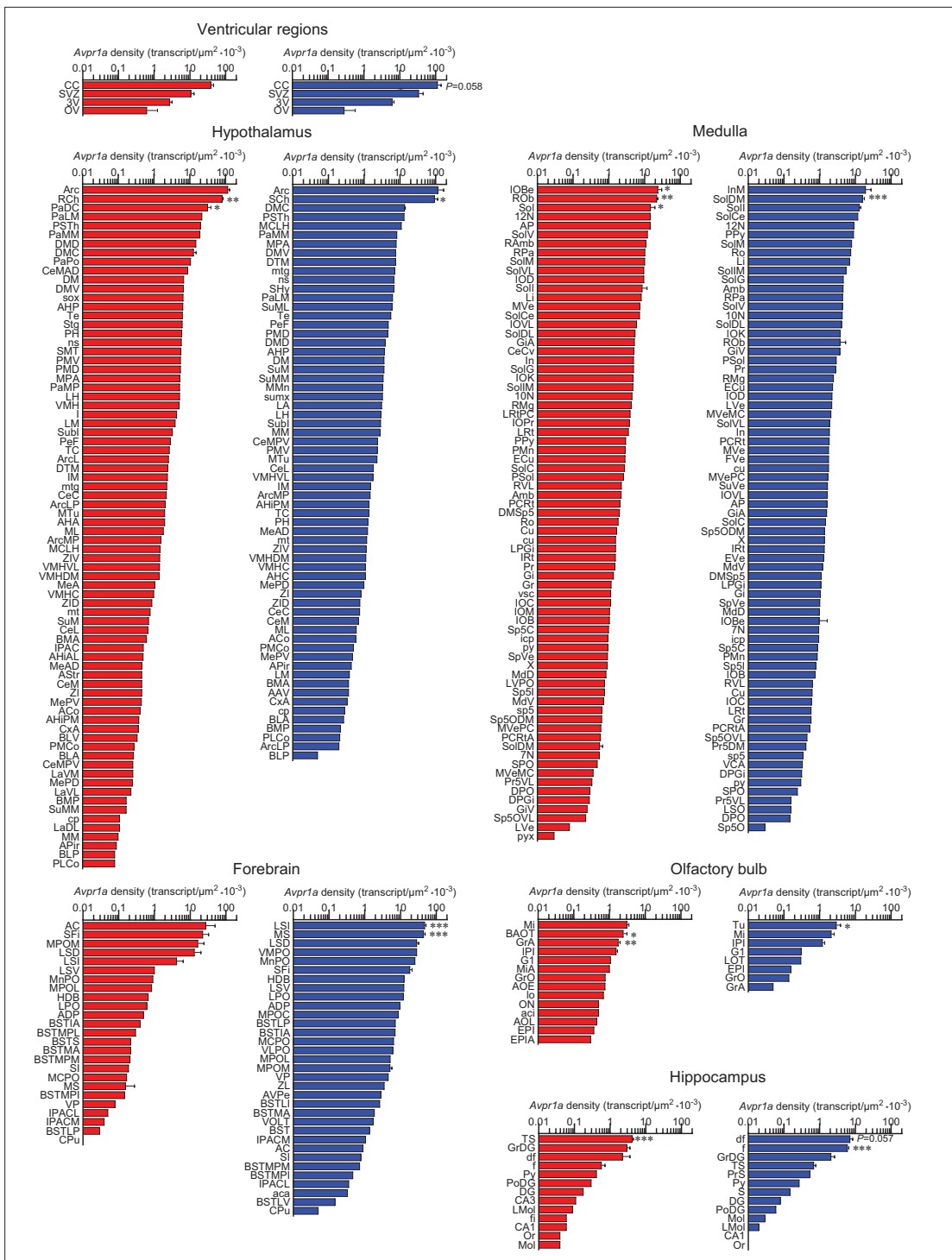

**Figure 3.** Sex-specific *Avpr1a* transcript density in the brain. *Avpr1a* transcript density in nuclei, subnuclei, and regions of the ventricular regions, hypothalamus, medulla, forebrain, olfactory bulb, and hippocampus. N=3 mice *per* sex, values are shown as means ± SE. ***p<0.001, **p<0.01, and *p<0.05, unpaired Student's *t* test and two-way ANOVA.

The online version of this article includes the following figure supplement(s) for figure 3:

**Figure supplement 1.** Sex-specific *Avpr1a* transcript numbers in the brain.

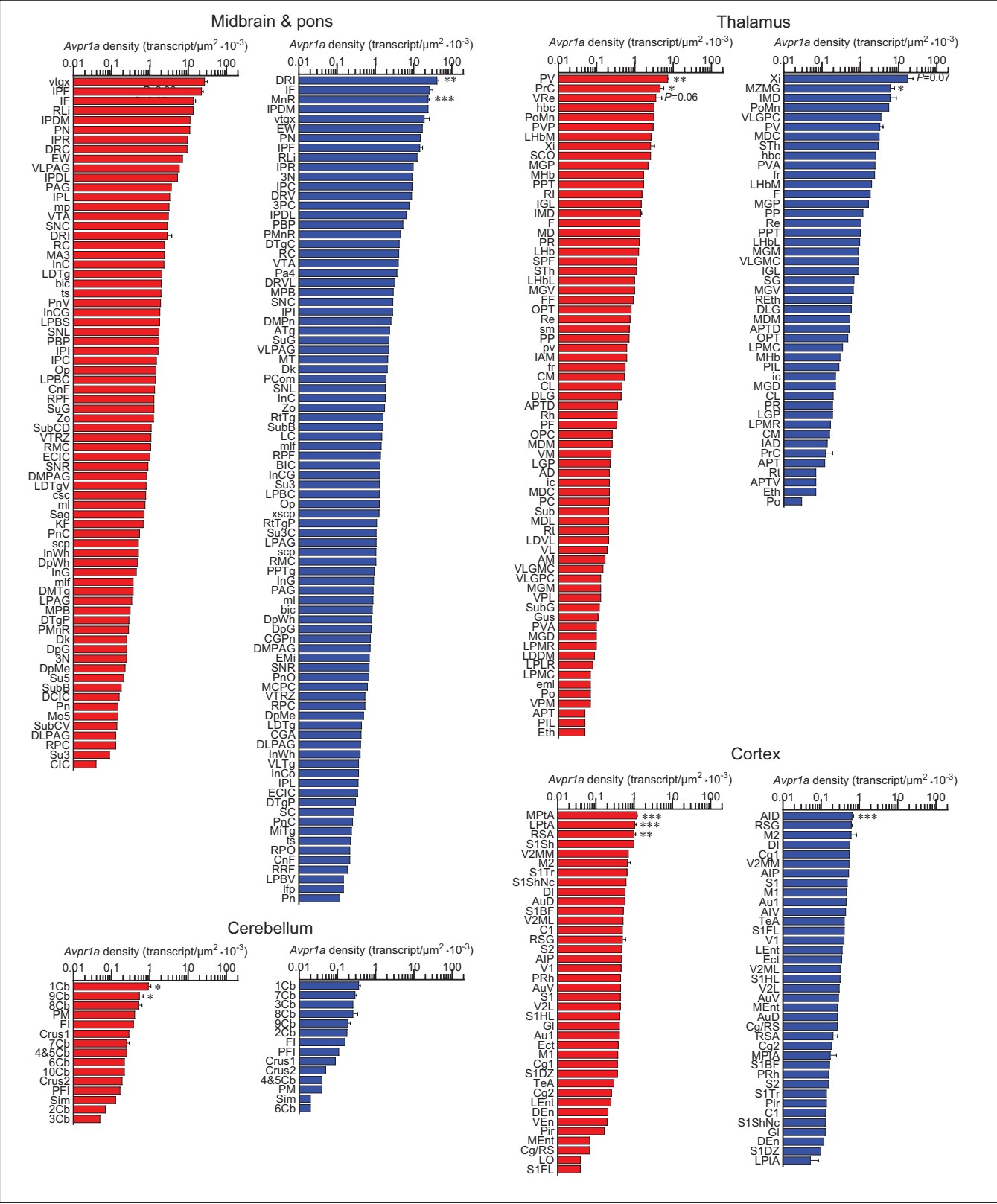

**Figure 4.** Sex-specific *Avpr1a* transcript density in the brain. *Avpr1a* transcript density in nuclei, subnuclei, and regions of the midbrain and pons, thalamus, cortex, and cerebellum. N=3 mice *per* sex, values are shown as means ± SE. ***p<0.001, **p<0.01, and *p<0.05, unpaired Student's *t* test and two-way ANOVA.

*Figure 4 continued on next page*

*Figure 4 continued*

The online version of this article includes the following figure supplement(s) for figure 4:

**Figure supplement 1.** Sex-specific *Avpr1a* transcript numbers in the brain.

their lives, has rewarding properties fundamental to maternal behavior and survival. Modulation of AVPR1a in the dorsal raphe nucleus has also been linked to social and emotional behaviors (*Dumais and Veenema, 2016*; *Rigney et al., 2020*; *Rood et al., 2013*). Sexual dimorphism in AVP innervation of and AVPR1a expression in the DRI appears to imprint dimorphic social behaviors. That is, AVPR1a blockade in the lateral habenular nucleus (LHb) of males, but not females who have lesser AVP innervation of the LHb and dorsal raphe nucleus, results in reduced urine marking to unfamiliar males and ultrasonic vocalizations to unfamiliar, sexually receptive females, whereas AVPR1a blockade in the dorsal raphe nucleus of only males reduces urine marking to unfamiliar males (*Rigney et al., 2020*).

In the forebrain, both sexes displayed high *Avpr1a* transcript density in septal nuclei. The highest *Avpr1a* density was in the anterior commissural nucleus (AC) within the septal nuclei of females, and lateral (LSI) and medial (MS) septal nuclei of males. It is not surprising that *Avpr1a* transcript density was significantly greater in septal nuclei of males than females, given that in many rodent species males have more AVP-immunoreactive fibers in the lateral septum (*de Vries et al., 1981*). Notably, the effects of AVP on social recognition is mediated via AVPR1a in the lateral septum (*Veenema et al., 2012*; *Bielsky et al., 2005*). For example, AVPR1a blockade inhibits social recognition in the rat, while AVPR1a knockout mice fail to display social recognition (*Veenema et al., 2012*; *Bielsky et al., 2004*; *Bielsky et al., 2005*).

Despite *Avpr1a* transcript densities in other brain divisions being significantly lower, sexually dimorphic differences are worth mentioning here. In the olfactory bulb, females had the high *Avpr1a* density in the mitral cell layer (Mi), whereas males had high receptor expression in the olfactory tubercle (Tu). A population of AVP neurons in the olfactory bulb of the rat that plays a role in social recognition via AVPR1a has been reported (*Tobin et al., 2010*). Silencing the AVPR1a by siRNA impairs habituation/dishabituation to juvenile cues, but not to volatile odors (*Tobin et al., 2010*). Of note, AVP is a retrograde signal that filters activation of the Mi cells in the ewe, likely through presynaptic modulation of norepinephrine or acetylcholine. The secretion of both transmitters is stimulated by AVP in the olfactory bulb (*Tobin et al., 2010*; *Lévy et al., 1995*). The functional relevance of AVP signaling via AVPR1a activation in the Tu requires additional studies. There is, however, evidence in the rat that AVP via AVPR1a has, at least, an indirect impact on Tu function, as seen by a reduction in activation responding to a noxious odor of butyric acid, when the AVPR1a is blocked (*Reed et al., 2013*). The presence of AVPR1a in the hippocampus, thalamus, cortex, and ventricular regions is consistent with the reported effects of AVP on memory (*de Wied, 1971*), emotional and reward-motivated behavior (*Zhang et al., 2006*), blood pressure (*Matsuguchi et al., 1982*), blood flow, and CSF production (*Faraci et al., 1988*). Functional roles of many other nuclei shown here to express AVPR1a and not mentioned in this report are much less clear. The importance of revealing novel AVP-triggered functions by interrogating AVPR1a site-specifically will require further investigations.

Collectively, our results provide compelling evidence of distinct and novel AVP/AVPR1a neuronal nodes in the brain. While studies on central AVP signaling and its control of blood pressure, water balance, and diverse social behaviors in mammals occupy the vast majority of the literature (*Silva et al., 1969*; *Stockand et al., 2022*; *Lukas and Neumann, 2013*; *Lukas et al., 2013*; *Veenema and Neumann, 2008*; *Meyer-Lindenberg et al., 2011*; *Albers, 2015*), we expect that this comprehensive compendium of sex-specific AVP/AVPR1a expression in the brain will deepen our understanding of the functional and neuroanatomical basis underlying old and new paradigm-shifting functions of central AVP signaling. As appears to be the case for most brain areas, the original discovery of function tends to become dogma, thereby leading to an oversimplification of multiple functions of those brain areas as they interact in circuits. Finally, the approach of direct mapping of receptor expression in the brain and periphery provides the groundwork for greater discernment of new functional arrangements of ancient pituitary glycoprotein hormones and nonapeptides, such as AVP and OXT, and provides helpful pointers toward improving pharmacological interventions in disease.

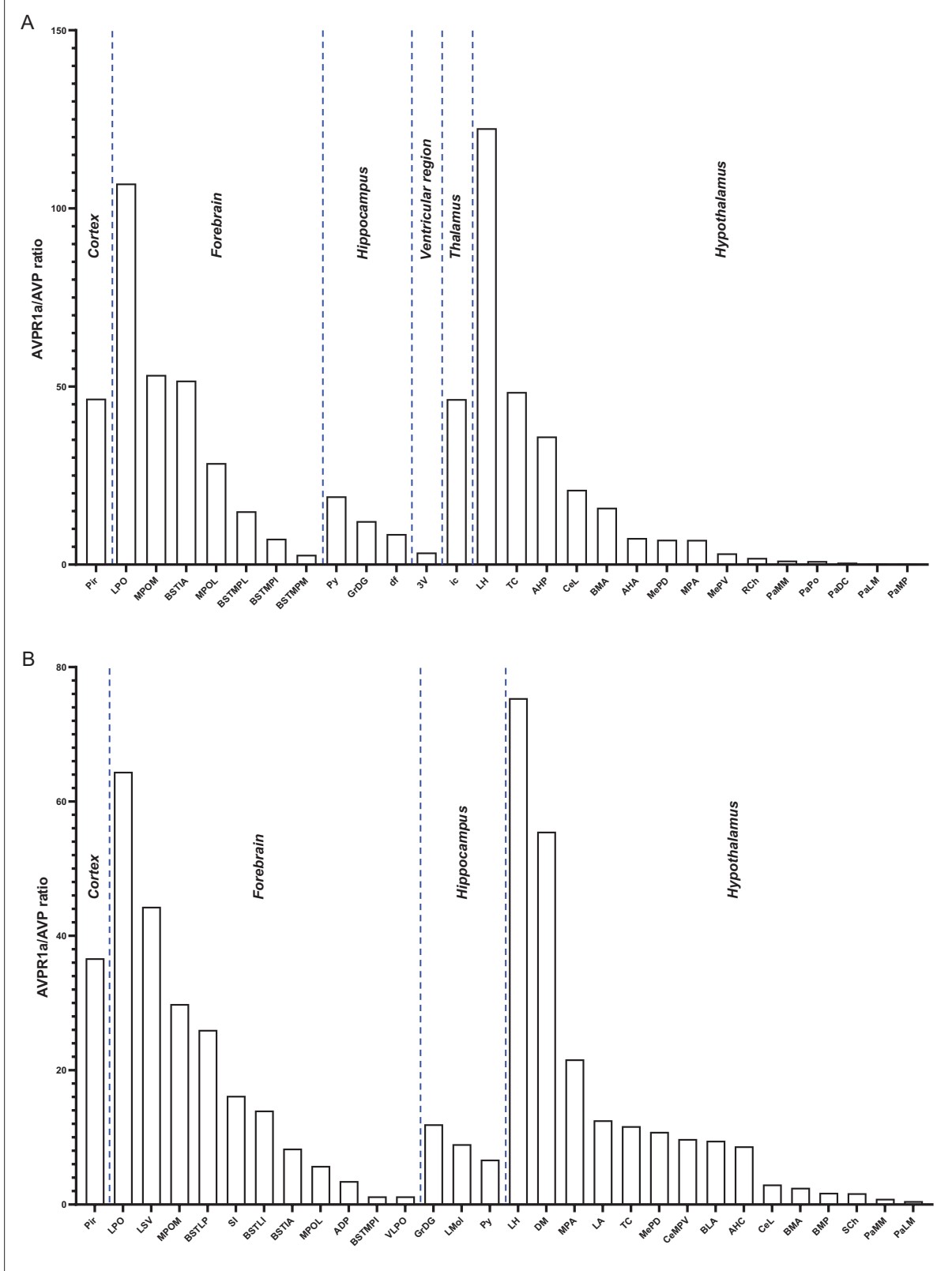

**Figure 5.** *Avp* and *Avpr1a* co-localization in the brain. We found that various nuclei and subnuclei exhibited *Avp* and *Avpr1a* co-localization in the brain of both sexes. (**A**) Female and (**B**) male *Avpr1a* to *Avp* ratios in the cortex, forebrain, hippocampus, 3rd ventricular region, thalamus, and hypothalamus.

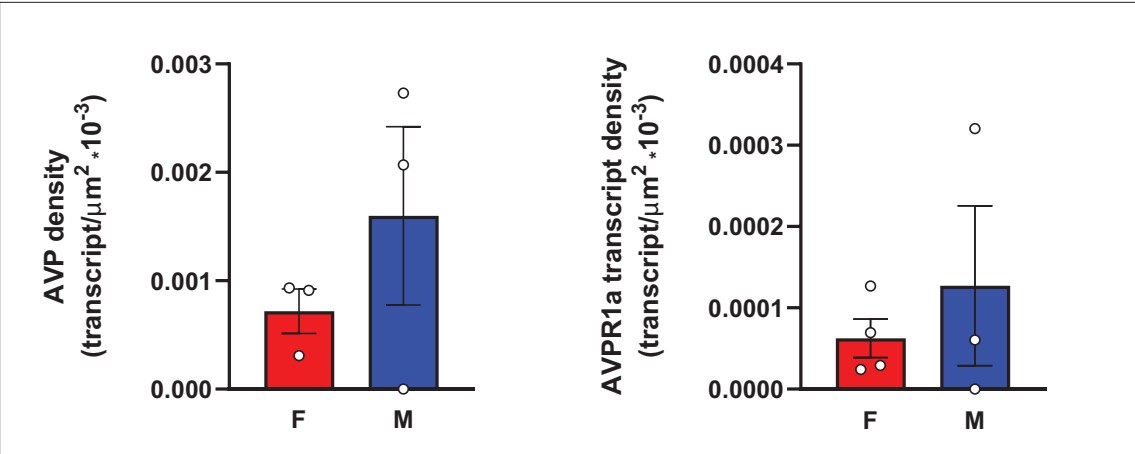

**Figure 6.** *Avp* and *Avpr1a* transcript densities in the pituitary gland. RNAscope revealed *Avp* and *Avpr1a* expression in the posterior pituitary lobe with *Avp* and *Avpr1a* transcript densities that were higher in male compared with female mice. N=3 mice *per* sex, values are shown as means ± SE. Unpaired Student's *t* test.

## Methods

### Mice

Adult C57BL/6J mice (~3- to 4-month-old) were housed in a 12 hr:12 hr light:dark cycle at 22 ± 2°C with ad libitum access to water and regular chow. All procedures were approved by the Mount Sinai Institutional Animal Care and Use Committee and are in accordance with Public Health Service and United States Department of Agriculture guidelines. Ethical approval for all experimental procedures was obtained from the appropriate Institutional Review Board under protocol number PROTO202100038.

### RNAscope

For RNAscope, mice were anesthetized with isoflurane (2–3% in oxygen; Baxter Healthcare, Deerfield, IL, USA) and transcardially perfused with 0.9% heparinized saline followed by 10% Neutral Buffered Formalin (NBF). Brains were promptly extracted, post-fixed in 10% NBF for 24 hr, dehydrated, and paraffin-embedded. Coronal sections were cut at 5 μm, with every tenth section mounted onto ~60 slides with 3 sections on each slide. This method allows covering the entire brain and eliminates the likelihood of counting the same transcript twice. Sections were air-dried overnight at room temperature and stored at 4°C until required.

Detection of mouse *Avp* and *Avpr* (*Avpr1a*) was performed separately on paraffin sections using Advanced Cell Diagnostics (ACD) RNAscope 2.5 LS Reagent Kit (#322100) and two RNAscope 2.5 LS probes, namely Mm-AVP-O1 (#472268) and Mm-AVPR1a (#418068). The kidney and prefrontal cortex served as positive and negative controls for AVPR1a, respectively. As with AVP, magnocellular cells of the PVH and SON served as positive controls, while the brain from the AVP knockout mouse served as a negative control.

Slides were baked at 60°C for 1 hr, deparaffinized, incubated with hydrogen peroxide for 10 min at room temperature, pretreated with Target Retrieval Reagent (#322001) for 20 min at 100°C and then with Protease III for 30 min at 40°C. Probe hybridization and signal amplification were performed as per the manufacturer's instructions for chromogenic assays.

Following the RNAscope assay, the slides were scanned at ×20 magnification, and the digital image analysis was successfully validated using the CaseViewer 2.4 software (3DHISTECH). The same software was employed to capture and prepare images for the figures in the article. Images of control tissues were taken using the microscope Leica DM 1000 LED. Detection of *Avp*- and *Avpr1a*-positive cells was also performed using the QuPath-0.2.3 (University of Edinburgh, UK) software. *The Atlas for the Mouse Brain in Stereotaxic Coordinates* (**Paxinos and Franklin, 2007**) was utilized to identify and manually map every nucleus, subnucleus, or region using the drawing features of the QuPath-0.2.3 software in every tenth brain section. This was followed by exhaustive counting of *Avp* and *Avpr1a* transcripts using a tag feature. *Avp* and *Avpr1a* transcript density was calculated by dividing the

absolute numbers by the total area (µm², ImageJ) of every nucleus, subnucleus, or region. Photomicrographs were prepared using Photoshop CS5 (Adobe Systems) only to adjust brightness, contrast, and sharpness, to remove artifacts (e.g., obscuring bubbles) and to make composite plates.

## Quantitation, validation, and statistical analysis

Data were analyzed by Student's *t*-test and two-way analysis of variance (ANOVA) followed by Tukey's multiple comparisons tests using GraphPad Prism 10.2.2 version (La Jolla, CA, USA). Significance was set at $p < 0.05$. p-values are shown.

## Acknowledgements

Work at Icahn School of Medicine at Mount Sinai carried at the Center for Translational Medicine and Pharmacology was supported by R01 AG071870 to MZ, TY, and S-MK; R01 AG074092 and U01AG073148 to TY and MZ; U19 AG060917 and R01 DK113627 to MZ.

## Additional information

### Competing interests

Weibin Zhou: Reviewing editor, *eLife*. The other authors declare that no competing interests exist.

### Funding

| Funder | Grant reference number | Author |
| --- | --- | --- |
| National Institutes of Health | AG071870 | Se-Min Kim<br>Tony Yuen<br>Mone Zaidi |
| National Institutes of Health | AG073148 | Tony Yuen<br>Mone Zaidi |
| National Institutes of Health | AG074092 | Mone Zaidi<br>Tony Yuen |
| National Institutes of Health | U19 AG060917 | Mone Zaidi |
| National Institutes of Health | DK113627 | Mone Zaidi |

The funders had no role in study design, data collection and interpretation, or the decision to submit the work for publication.

### Author contributions

Anisa Azatovna Gumerova, Data curation, Writing – review and editing; Georgii Pevnev, Data curation, Formal analysis; Funda Korkmaz, Guzel Burganova, Victoria Laurencin, Investigation; Uliana Cheliadinova, Data curation, Investigation; Darya Vasilyeva, Software, Investigation; Liam Cullen, Data curation; Orly Barak, Resources; Farhath Sultana, Software, Validation; Weibin Zhou, Steven Lee Sims, Validation; Emily Weiss, Writing – review and editing; Tal Frolinger, Ki A Goosens, Validation, Visualization; Se-Min Kim, Data curation, Validation; Tony Yuen, Supervision, Funding acquisition, Validation, Visualization, Writing – review and editing; Mone Zaidi, Conceptualization, Data curation, Supervision, Funding acquisition, Validation, Writing – review and editing; Vitaly Ryu, Conceptualization, Data curation, Formal analysis, Supervision, Writing - original draft, Writing – review and editing

### Author ORCIDs

Anisa Azatovna Gumerova ⓘ https://orcid.org/0000-0002-1449-4000
Georgii Pevnev ⓘ https://orcid.org/0000-0003-2015-9310
Funda Korkmaz ⓘ https://orcid.org/0000-0002-9174-8369
Uliana Cheliadinova ⓘ https://orcid.org/0009-0007-2308-4824
Guzel Burganova ⓘ https://orcid.org/0000-0002-7204-7268
Darya Vasilyeva ⓘ https://orcid.org/0009-0007-5121-9618

Orly Barak https://orcid.org/0000-0002-3041-2343
Farhath Sultana https://orcid.org/0000-0003-4186-3390
Steven Lee Sims https://orcid.org/0000-0002-1636-084X
Victoria Laurencin https://orcid.org/0009-0006-0801-6471
Tal Frolinger https://orcid.org/0000-0002-2066-6649
Ki A Goosens https://orcid.org/0000-0002-5246-2261
Mone Zaidi https://orcid.org/0000-0001-5911-9522
Vitaly Ryu https://orcid.org/0000-0001-8068-4577

Reviewer #1 (Public review): https://doi.org/10.7554/eLife.105355.4.sa1
Reviewer #2 (Public review): https://doi.org/10.7554/eLife.105355.4.sa2
Author response https://doi.org/10.7554/eLife.105355.4.sa3

## Additional files

### Supplementary files
MDAR checklist

### Data availability
The authors uploaded the dataset in Dryad to maintain high standards of research reproducibility.

The following dataset was generated:

| Author(s) | Year | Dataset title | Dataset URL | Database and Identifier |
|---|---|---|---|---|
| Gumerova AA, Pevnev G, Korkmaz F, Cheliadinova U, Burganova G, Vasilyeva D, Cullen L, Barak O, Sultana F, Zhou W, Sims SL, Weiss E, Laurencin V, Frolinger T, Kim S, Goosens KA, Yuen T, Zaidi M, Ryu V | 2026 | Sex-specific Single Transcript Level Atlas of Vasopressin and its Receptor (AVPR1a) in the Mouse Brain | http://doi.org/10.5061/dryad.np5hqc08h | Dryad Digital Repository, 10.5061/dryad.np5hqc08h |

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

# Appendix 1

**Appendix 1—table 1.** Glossary of the brain nuclei, sub–nuclei and regions.

**Olfactory bulb**

| | |
|---|---|
| aci | anterior commissure, intrabulbar part |
| AOE | anterior olfactory nucleus, external part |
| AOL | anterior olfactory nucleus, lateral part |
| BAOT | bed nucleus of the accessory olfactory tract |
| EPl | external plexiform layer of the olfactory bulb |
| EPlA | external plexiform layer of the accessory olfactory bulb |
| G1 | glomerular layer of the olfactory bulb |
| GrA | granule cell layer of the accessory olfactory bulb |
| GrO | granular cell layer of the olfactory bulb |
| IPI | interpeduncular nucleus, intermediate subnucleus |
| lo | lateral olfactory tract |
| LOT | nucleus of the lateral olfactory tract |
| Mi | mitral cell layer of the olfactory bulb |
| MiA | mitral cell layer of the accessory olfactory bulb |
| ON | olfactory nerve layer |
| Tu | olfactory tubercle |

**Cerebral cortex**

| | |
|---|---|
| AID | agranular insular cortex, dorsal part |
| AIP | agranular insular cortex, posterior part |
| AIV | agranular insular cortex, ventral part |
| Au1 | primary auditory cortex |
| AuD | secondary auditory cortex, dorsal area |
| AuV | secondary auditory cortex, ventral area |
| Cg/RS | cingular/retrosplenial cortex |
| Cg1 | cingulate cortex, area 1 |
| Cg2 | cingulate cortex, area 2 |
| C1 | C1 adrenaline cells |
| DEn | dorsal endopiriform nucleus |
| DI | dysgranular insular cortex |
| Ect | ectorhinal cortex |
| GI | granular insular cortex |
| LEnt | lateral entorhinal cortex |
| LO | lateral orbital cortex |
| LPtA | lateral parietal association cortex |
| M1 | primary motor cortex |
| M2 | secondary motor cortex |

*Continued on next page*

*Continued*

**Cerebral cortex**

| | |
|---|---|
| MEnt | medial entorhinal cortex |
| MPtA | medial parietal association cortex |
| Pir | piriform cortex |
| PRh | perirhinal cortex |
| RSA | retrosplenial agranular cortex |
| RSG | retrosplenial granular cortex |
| S1 | primary somatosensory cortex |
| S1BF | primary somatosensory cortex, barrel field |
| S1DZ | primary somatosensory cortex, dysgranular region |
| S1FL | primary somatosensory cortex, forelimb region |
| S1HL | primary somatosensory cortex, hindlimb region |
| S1Sh | primary somatosensory cortex, shoulder region |
| S1ShNc | primary somatosensory cortex, shoulder/neck region |
| S1Tr | primary somatosensory cortex, trunk region |
| S2 | secondary somatosensory cortex |
| TeA | temporal association cortex |
| V1 | primary visual cortex |
| V2L | secondary visual cortex, lateral area |
| V2ML | secondary visual cortex, mediolateral area |
| V2MM | secondary visual cortex, mediomedial area |
| VEn | ventral endopiriform nucleus |

**Forebrain**

| | |
|---|---|
| AC | anterior commissural nucleus |
| aca | anterior commissure, anterior part |
| ADP | anterodorsal preoptic nucleus |
| AVPe | anteroventral periventricular nucleus |
| BAC | bed nucleus of the anterior commissure |
| BST | bed nucleus of the stria terminalis |
| BSTIA | bed nucleus of the stria terminalis, intraamygdaloid division |
| BSTLI | bed nucleus of the stria terminalis, lateral division, intermediate part |
| BSTLP | bed nucleus of the stria terminalis, lateral division, posterior part |
| BSTLV | bed nucleus of the stria terminalis, lateral division, ventral part |
| BSTMA | bed nucleus of the stria terminalis, medial division, anterior part |
| BSTMPI | bed nucleus of the stria terminalis, medial division, posterointermediate part |
| BSTMPL | bed nucleus of the stria terminalis, medial division, posterolateral part |
| BSTMPM | bed nucleus of the stria terminalis, medial division, posteromedial part |
| BSTS | bed nucleus of stria terminalis, supracapsular part |
| CPu | caudate putamen (striatum) |

*Continued*

**Forebrain**

| | |
|---|---|
| HDB | nucleus of the horizontal limb of the diagonal band |
| IPACL | interstitial nucleus of the posterior limb of the anterior commissure, lateral part |
| IPACM | interstitial nucleus of the posterior limb of the anterior commissure, medial part |
| LPO | lateral preoptic area |
| LSD | lateral septal nucleus, dorsal part |
| LSI | lateral septal nucleus, intermediate part |
| LSV | lateral septal nucleus, ventral part |
| MCPO | magnocellular preoptic nucleus |
| MnPO | median preoptic nucleus |
| MPOC | medial preoptic nucleus, central part |
| MPOL | medial preoptic nucleus, lateral part |
| MPOM | medial preoptic nucleus, medial part |
| MS | medial septal nucleus |
| SFi | septofimbrial nucleus |
| SI | substantia innominata |
| VLPO | ventrolateral preoptic nucleus |
| VMPO | ventromedial preoptic nucleus |
| VOLT | vascular organ of the lamina terminalis |
| VP | ventral pallidum |
| ZL | zona limitans |

**Hippocampus**

| | |
|---|---|
| CA1 | field ca1 of hippocampus |
| CA3 | field CA3 of hippocampus |
| df | dorsal fornix |
| DG | dentate gyrus |
| dhc | dorsal hippocampal commissure |
| f | fornix |
| fi | fimbria of the hippocampus |
| GrDG | granular layer of the dentate gyrus |
| LMol | lacunosum moleculare layer of the hippocampus |
| Mol | molecular layer of the dentate gyrus |
| Or | oriens layer of the hippocampus |
| PoDG | polymorph layer of the dentate gyrus |
| PrS | presubiculum |
| Py | pyramidal tract |
| S | subiculum |
| TS | triangular septal nucleus |

**Thalamus**

*Continued on next page*

*Continued*

**Hippocampus**

| | |
|---|---|
| AD | anterodorsal thalamic nucleus |
| AM | anteromedial thalamic nucleus |
| APT | anterior pretectal nucleus |
| APTD | anterior pretectal nucleus, dorsal part |
| APTV | anterior pretectal nucleus, ventral part |
| CL | centrolateral thalamic nucleus |
| CM | central medial thalamic nucleus |
| DLG | dorsal lateral geniculate nucleus |
| eml | external medullary lamina |
| Eth | ethmoid thalamic nucleus |
| F | nucleus of the fields of Forel |
| FF | fields of Forel |
| fr | fasciculus retroflexus |
| Gus | gustatory thalamic nucleus |
| hbc | habenular commissure |
| IAD | interanterodorsal thalamic nucleus |
| IAM | interanteromedial thalamic nucleus |
| ic | internal capsule |
| IGL | intergeniculate leaf |
| IMD | intermediodorsal thalamic nucleus |
| LDDM | laterodorsal thalamic nucleus, dorsomedial part |
| LDVL | laterodorsal thalamic nucleus, ventrolateral part |
| LGP | lateral globus pallidus |
| LHb | lateral habenular nucleus |
| LHbL | lateral habenular nucleus, lateral part |
| LHbM | lateral habenular nucleus, medial part |
| LPLR | lateral posterior thalamic nucleus, laterorostral part |
| LPMC | lateral posterior thalamic nucleus, mediocaudal part |
| LPMR | lateral posterior thalamic nucleus, mediorostral part |
| MD | mediodorsal thalamic nucleus |
| MDC | mediodorsal thalamic nucleus, central part |
| MDL | mediodorsal thalamic nucleus, lateral part |
| MDM | mediodorsal thalamic nucleus, medial part |
| MGD | medial geniculate nucleus, dorsal part |
| MGM | medial geniculate nucleus, medial part |
| MGP | medial globus pallidus (entopeduncular nucleus) |
| MGV | medial geniculate nucleus, ventral part |
| MHb | medial habenular nucleus |

*Continued*

**Hippocampus**

| | |
|---|---|
| MZMG | marginal zone of the medial geniculate |
| OPC | oval paracentral thalamic nucleus |
| OPT | olivary pretectal nucleus |
| PC | paracentral thalamic nucleus |
| PF | parafascicular thalamic nucleus |
| PIL | posterior intralaminar thalamic nucleus |
| Po | posterior thalamic nuclear group |
| PoMn | posteromedian thalamic nucleus |
| PP | peripeduncular nucleus |
| PPT | posterior pretectal nucleus |
| PR | prerubral field |
| PrC | precommissural nucleus |
| pv | periventricular fiber system |
| PV | paraventricular thalamic nucleus |
| PVA | paraventricular thalamic nucleus, anterior part |
| PVP | paraventricular thalamic nucleus, posterior part |
| Re | reuniens thalamic nucleus |
| REth | retroethmoid nucleus |
| Rh | rhomboid thalamic nucleus |
| RI | rostral interstitial nucleus of medial longitudinal fasciculus |
| Rt | reticular thalamic nucleus |
| SCO | subcommissural organ |
| SG | suprageniculate thalamic nucleus |
| sm | stria medullaris of the thalamus |
| SPF | subparafascicular thalamic nucleus |
| STh | subthalamic nucleus |
| Sub | submedius thalamic nucleus |
| SubG | subgeniculate nucleus |
| VL | ventrolateral thalamic nucleus |
| VLGMC | ventral lateral geniculate nucleus, magnocellular part |
| VLGPC | ventral lateral geniculate nucleus, parvicellular part |
| VM | ventromedial thalamic nucleus |
| VPL | ventral posterolateral thalamic nucleus |
| VPM | ventral posteromedial thalamic nucleus |
| VRe | ventral reuniens thalamic nucleus |
| Xi | xiphoid thalamic nucleus |

**Hypothalamus**

| | |
|---|---|
| AAV | anterior amygdaloid area, ventral part |

*Continued on next page*

*Continued*

**Hypothalamus**

| | |
|---|---|
| ACo | anterior cortical amygdaloid nucleus |
| AHA | anterior hypothalamic area, anterior part |
| AHC | anterior hypothalamic area, central part |
| AHiAL | amygdalohippocampal area, anterolateral part |
| AHiPM | amygdalohippocampal area, posteromedial part |
| AHP | anterior hypothalamic area, posterior part |
| APir | amygdalopiriform transition area |
| Arc | arcuate hypothalamic nucleus |
| ArcL | arcuate hypothalamic nucleus, lateral part |
| ArcLP | arcuate hypothalamic nucleus, lateroposterior part |
| ArcMP | arcuate hypothalamic nucleus, medial posterior part |
| AStr | amygdalostriatal transition area |
| BLA | basolateral amygdaloid nucleus, anterior part |
| BLP | basolateral amygdaloid nucleus, posterior part |
| BLV | basolateral amygdaloid nucleus, ventral part |
| BMA | basolateral amygdaloid nucleus, anterior part |
| BMP | basomedial amygdaloid nucleus, posterior part |
| CeC | central amygdaloid nucleus, capsular part |
| CeL | central amygdaloid nucleus, lateral division |
| CeM | central amygdaloid nucleus, medial division |
| CeMAD | central amygdaloid nucleus, medial division, anterodorsal part |
| CeMPV | central amygdaloid nucleus, medial posteroventral part |
| cp | cerebral peduncle, basal part |
| CxA | cortex-amygdala transition zone |
| DM | dorsomedial hypothalamic nucleus |
| DMC | dorsomedial hypothalamic nucleus, compact part |
| DMD | dorsomedial hypothalamic nucleus, dorsal part |
| DMV | dorsomedial hypothalamic nucleus, ventral part |
| DTM | dorsal tuberomammillary nucleus |
| I | intercalated nuclei of the amygdala |
| IM | intercalated amygdaloid nucleus, main part |
| IPAC | interstitial nucleus of the posterior limb of the anterior commissure |
| LA | lateroanterior hypothalamic nucleus |
| LaDL | lateral amygdaloid nucleus, dorsolateral part |
| LaVL | lateral amygdaloid nucleus, ventrolateral part |
| LaVM | lateral amygdaloid nucleus, ventromedial part |
| LH | lateral hypothalamic area |
| LM | lateral mammillary nucleus |

*Continued*

**Hypothalamus**

| | |
|---|---|
| MCLH | magnocellular nucleus of the lateral hypothalamus |
| MeA | medial amygdaloid nucleus, anterior part |
| MeAD | medial amygdaloid nucleus, anteriodorsal part |
| MePD | medial amygdaloid nucleus, posterodorsal part |
| MePV | medial amygdaloid nucleus, posteroventral part |
| ML | medial mammillary nucleus, lateral part |
| MM | medial mammillary nucleus, medial part |
| MMn | medial mammillary nucleus, median part |
| MPA | medial preoptic area |
| mt | mammillothalamic tract |
| mtg | mammillotegmental tract |
| MTu | medial tuberal nucleus |
| ns | nigrostriatal bundle |
| opt | optic tract |
| PaAP | paraventricular hypothalamic nucleus, anterior parvicellular part |
| PaDC | paraventricular hypothalamic nucleus, dorsal cap |
| PaLM | paraventricular hypothalamic nucleus, lateral magnocellular part |
| PaMM | paraventricular hypothalamic nucleus, medial magnocellular part |
| PaMP | paraventricular hypothalamic nucleus, medial parvicellular part |
| PaPo | paraventricular hypothalamic nucleus, posterior part |
| PaV | paraventricular hypothalamic nucleus, ventral part |
| Pe | periventricular hypothalamic nucleus |
| PeF | perifornical nucleus |
| PH | posterior hypothalamic area |
| PLCo | posterolateral cortical amygdaloid nucleus |
| PMCo | posteromedial cortical amygdaloid nucleus (C3) |
| PMD | premammillary nucleus, dorsal part |
| PMV | premammillary nucleus, ventral part |
| PSTh | parasubthalamic nucleus |
| RCh | retrochiasmatic area |
| SCh | suprachiasmatic nucleus |
| SChDM | suprachiasmatic nucleus, dorsomedial part |
| SChVL | suprachiasmatic nucleus, ventrolateral part |
| SHy | septohypothalamic nucleus |
| SMT | submammillothalamic nucleus |
| SO | supraoptic nucleus |
| SOR | supraoptic nucleus, retrochiasmatic part |
| sox | supraoptic decussation |

*Continued*

**Hypothalamus**

| | |
|---|---|
| SPa | subparaventricular zone of the hypothalamus |
| Stg | stigmoid hypothalamic nucleus |
| Subl | subincertal nucleus |
| SuM | supramammillary nucleus |
| SuML | supramammillary nucleus, lateral part |
| SuMM | supramammillary nucleus, medial part |
| sumx | supramammillary decussation |
| TC | tuber cinereum area |
| Te | terete hypothalamic nucleus |
| VMH | ventromedial hypothalamic nucleus |
| VMHC | ventromedial hypothalamic nucleus, central part |
| VMHDM | ventromedial hypothalamic nucleus, dorsomedial part |
| VMHVL | ventromedial hypothalamic nucleus, ventrolateral part |
| ZI | zona incerta |
| ZID | zona incerta, dorsal part |
| ZIV | zona incerta, ventral part |

**Midbrain and pons**

| | |
|---|---|
| 3 N | oculomotor nucleus |
| 3PC | oculomotor nucleus, parvicellular part |
| ATg | anterior tegmental nucleus |
| bic | brachium of the inferior colliculus |
| BIC | nucleus of the brachium of the inferior colliculus |
| CGA | central gray, alpha part |
| CGPn | central gray of the pons |
| CIC | central nucleus of the inferior colliculus |
| CnF | cuneiform nucleus |
| csc | commissure of the superior colliculus |
| DCIC | dorsal cortex of the inferior colliculus |
| Dk | nucleus of Darkschewitsch |
| DLPAG | dorsolateral periaqueductal gray |
| DMPAG | dorsomedial periaqueductal gray |
| DMPn | dorsomedial pontine nucleus |
| DMTg | dorsomedial tegmental area |
| DpG | deep gray layer of the superior colliculus |
| DpMe | deep mesencephalic nucleus |
| DpWh | deep white layer of the superior colliculus |
| DRC | dorsal raphe nucleus, caudal part |
| DRI | dorsal raphe nucleus, interfascicular part |

*Continued*

**Midbrain and pons**

| | |
|---|---|
| DRV | dorsal raphe nucleus, ventral part |
| DRVL | dorsal raphe nucleus, ventrolateral part |
| DTgC | dorsal tegmental nucleus, central part |
| DTgP | dorsal tegmental nucleus, pericentral part |
| ECIC | external cortex of the inferior colliculus |
| EMi | epimicrocellular nucleus |
| EW | Edinger-Westphal nucleus |
| IF | interfascicular nucleus |
| InC | interstitial nucleus of Cajal |
| InCG | interstitial nucleus of Cajal, greater part |
| InCo | intercollicular nucleus |
| InG | intermediate gray layer of the superior colliculus |
| InWh | intermediate white layer of the superior colliculus |
| IPC | interpeduncular nucleus, caudal subnucleus |
| IPDL | interpeduncular nucleus, dorsolateral subnucleus |
| IPDM | interpeduncular nucleus, dorsomedial subnucleus |
| IPF | interpeduncular fossa |
| IPI | interpeduncular nucleus, intermediate subnucleus |
| IPL | interpeduncular nucleus, lateral subnucleus |
| IPR | interpeduncular nucleus, rostral subnucleus |
| KF | Ko¨lliker-Fuse nucleus |
| LC | locus coeruleus |
| LDTg | laterodorsal tegmental nucleus |
| LDTgV | laterodorsal tegmental nucleus, ventral part |
| lfp | longitudinal fasciculus of the pons |
| LPAG | lateral periaqueductal gray |
| LPBC | lateral parabrachial nucleus, central part |
| LPBS | lateral parabrachial nucleus, superior part |
| LPBV | lateral parabrachial nucleus, ventral part |
| MA3 | medial accessory oculomotor nucleus |
| MCPC | magnocellular nucleus of the posterior commissure |
| MiTg | microcellular tegmental nucleus |
| ml | medial lemniscus |
| mlf | medial longitudinal fasciculus |
| MnR | median raphe nucleus |
| Mo5 | motor trigeminal nucleus |
| mp | mammillary peduncle |
| MPB | medial parabrachial nucleus |

*Continued*

**Midbrain and pons**

| | |
|---|---|
| MT | medial terminal nucleus of the accessory optic tract |
| Op | optic nerve layer of the superior colliculus |
| Pa4 | paratrochlear nucleus |
| PAG | periaqueductal gray |
| PBP | parabrachial pigmented nucleus |
| PCom | nucleus of the posterior commissure |
| PMnR | paramedian raphe nucleus |
| Pn | pontine nuclei |
| PN | paranigral nucleus |
| PnC | pontine reticular nucleus, caudal part |
| PnO | pontine reticular nucleus, oral part |
| PnV | pontine reticular nucleus, ventral part |
| PPTg | pedunculopontine tegmental nucleus |
| RC | raphe cap |
| RLi | rostral linear nucleus of the raphe |
| RMC | red nucleus, magnocellular part |
| RPC | red nucleus, parvicellular part |
| RPF | retroparafascicular nucleus |
| RPO | rostral periolivary region |
| RRF | retrorubral field |
| RtTg | reticulotegmental nucleus of the pons |
| RtTgP | reticulotegmental nucleus of the pons, pericentral part |
| Sag | sagulum nucleus |
| SC | superior colliculus |
| scp | superior cerebellar peduncle (brachium conjunctivum) |
| SNC | substantia nigra, compact part |
| SNL | substantia nigra, lateral part |
| SNR | substantia nigra, reticular part |
| Su3 | supraoculomotor periaqueductal gray |
| Su3C | supraoculomotor cap |
| SubB | subbrachial nucleus |
| SuG | superficial gray layer of the superior colliculus |
| ts | tectospinal tract |
| VLPAG | ventrolateral periaqueductal gray |
| VLTg | ventrolateral tegmental area |
| VTA | ventral tegmental area |
| vtgx | ventral tegmental decussation |
| VTRZ | visual tegmental relay zone |
| xscp | decussation of the superior cerebellar peduncle |

**Midbrain and pons**

| | |
|---|---|
| Zo | zonal layer of the superior colliculus |

**Medulla**

| | |
|---|---|
| 7 N | facial nucleus |
| 10 N | dorsal motor nucleus of vagus |
| 12 N | hypoglossal nucleus |
| Amb | ambiguus nucleus |
| AP | area postrema |
| CeCv | central cervical nucleus |
| cu | cuneate fasciculus |
| Cu | cuneate nucleus |
| DMSp5 | dorsomedial spinal trigeminal nucleus |
| DPGi | dorsal paragigantocellular nucleus |
| DPO | dorsal periolivary region |
| ECu | external cuneate nucleus |
| EVe | nucleus of origin of efferents of the vestibular nerve |
| FVe | F cell group of the vestibular complex |
| Gi | gigantocellular reticular nucleus |
| GiA | gigantocellular reticular nucleus, alpha part |
| GiV | gigantocellular reticular nucleus, ventral part |
| Gr | gracile nucleus |
| icp | inferior cerebellar peduncle (restiform body) |
| In | intercalated nucleus of the medulla |
| InM | intermedius nucleus of the medulla |
| IOB | inferior olive, subnucleus B of medial nucleus |
| IOBe | inferior olive, subnucleus B of medial nucleus |
| IOC | inferior olive, subnucleus C of medial nucleus |
| IOD | inferior olive, dorsal nucleus |
| IOK | inferior olive, cap of Kooy of the medial nucleus |
| IOVL | inferior olive, ventrolateral protrusion |
| IRt | intermediate reticular nucleus |
| Li | linear nucleus of the medulla |
| LPGi | lateral paragigantocellular nucleus |
| LRt | lateral reticular nucleus |
| LRtPC | lateral reticular nucleus, parvicellular part |
| LSO | lateral superior olive |
| LVe | lateral vestibular nucleus |
| LVPO | lateroventral periolivary nucleus |
| MdD | medullary reticular nucleus, dorsal part |
| MdV | medullary reticular nucleus, ventral part |

*Continued on next page*

*Continued*

**Medulla**

| | |
|---|---|
| MVe | medial vestibular nucleus |
| MVeMC | medial vestibular nucleus, magnocellular part |
| MVePC | medial vestibular nucleus, parvicellular part |
| PCRt | parvicellular reticular nucleus |
| PCRtA | parvicellular reticular nucleus, alpha part |
| PMn | paramedian reticular nucleus |
| PPy | parapyramidal nucleus |
| Pr | prepositus nucleus |
| Pr5DM | principal sensory trigeminal nucleus, dorsomedial part |
| Pr5VL | principal sensory trigeminal nucleus, ventrolateral part |
| PSol | parasolitary nucleus |
| py | pyramidal tract |
| pyx | pyramidal decussation |
| RAmb | retroambiguus nucleus |
| RMg | raphe magnus nucleus |
| Ro | nucleus of Roller |
| ROb | raphe obscurus nucleus |
| RPa | raphe pallidus nucleus |
| RVL | rostroventrolateral reticular nucleus |
| Sol | nucleus of the solitary tract |
| SolC | nucleus of the solitary tract, commissural part |
| SolCe | nucleus of the solitary tract, central part |
| SolDL | solitary nucleus, dorsolateral part |
| SolDM | nucleus of the solitary tract, dorsomedial part |
| SolG | nucleus of the solitary tract, gelatinous part |
| SolI | nucleus of the solitary tract, interstitial part |
| SolIM | nucleus of the solitary tract, intermediate part |
| SolM | nucleus of the solitary tract, medial part |
| SolV | solitary nucleus, ventral part |
| SolVL | nucleus of the solitary tract, ventrolateral part |
| sp5 | spinal trigeminal tract |
| Sp5C | spinal trigeminal nucleus, caudal part |
| Sp5I | spinal trigeminal nucleus, interpolar part |
| Sp5O | spinal trigeminal nucleus, oral part |
| Sp5ODM | spinal trigeminal nucleus, oral part, dorsomedial division |
| Sp5OVL | spinal trigeminal nucleus, oral part, ventrolateral division |
| SPO | superior paraolivary nucleus |
| SpVe | spinal vestibular nucleus |

*Continued*

**Medulla**

| | |
|---|---|
| SuVe | superior vestibular nucleus |
| VCA | ventral cochlear nucleus, anterior part |
| vsc | ventral spinocerebellar tract |
| X | nucleus X |

**Cerebellum**

| | |
|---|---|
| 1Cb | 1st Cerebellar lobule |
| 2Cb | 2nd Cerebellar lobule |
| 3Cb | 3rd Cerebellar lobule |
| 4&5Cb | 4&5th Cerebellar lobules |
| 6Cb | 6th Cerebellar lobule |
| 7Cb | 7th Cerebellar lobule |
| 8Cb | 8th Cerebellar lobule |
| 9Cb | 9th Cerebellar lobule |
| 10Cb | 10th Cerebellar lobule |
| Crus1 | crus 1 of the ansiform lobule |
| Crus2 | crus 2 of the ansiform lobule |
| Fl | flocculus |
| PFl | paraflocculus |
| PM | paramedian lobule |
| Sim | simple lobule |

**Ventricular zones**

| | |
|---|---|
| 3 V | 3rd ventricle |
| OV | olfactory ventricle (olfactory part of lateral ventricle) |
| OV | olfactory ventricle (olfactory part of lateral ventricle) |
| SVZ | subventricular zone |

