## [Editor Report · eLife Assessment]

This work presents a brain-wide atlas of vasopressin (Avp) and vasopressin receptor 1A (Avpr1a) mRNA expression in mouse brains using high-resolution RNAscope in situ hybridization. The single-transcript approach provides precise localization and identifies additional brain regions expressing Avpr1a, creating a **valuable** resource for the field. The revised manuscript is clearer and more impactful, with improved figures, stronger data organization, and enhanced scholarship through added context and citations. Overall, the evidence is **compelling**, and the atlas should be broadly of use to researchers studying vasopressin signaling and related neural circuits.

---

## [Referee Report · Reviewer #1 (Public review)]

Summary:

Despite accumulating prior studies on the expressions of AVP and AVPR1a in the brain, a detailed, gender-specific mapping of AVP/AVPR1a neuronal nodes has been lacking. Using RNAscope, a cutting-edge technology that detects single RNA transcripts, the authors created a comprehensive neuroanatomical atlas of Avp and Avpr1a in male and female brains.

Strengths:

This well-executed study provides valuable new insights into gender differences in the distribution of Avp and Avpr1a. The atlas is an important resource for the neuroscience community.

The authors have previously adequately addressed all of my concerns. I have no further questions or concerns.

---

## [Referee Report · Reviewer #2 (Public review)]

Summary:

The authors conducted a brain-wide survey of Avp (arginine vasopressin) and its Avpr1a gene expression in the mouse brain using RNAscope, a high-resolution in situ hybridization method. Overall, the findings are useful and important because they identify brain regions that express the Avpr1a transcript. A comprehensive overview of Avpr1a expression in the mouse brain could be highly informative and impactful. The authors used RNAscope (a proprietary in situ hybridization method) to assess transcript abundance of Avp and one of its receptors, Avpr1a. The finding of Avp-expressing cells outside the hypothalamus and the extended amygdala is novel and is nicely demonstrated by new photomicrographs in the revised manuscript. The Avpr1a data suggest expression in numerous brain regions. In the revised manuscript, reworked figures make the data easier to interpret.

Strengths:

A survey of Avpr1a expression in the mouse brain is an important tool for exploring vasopressin function in the mammalian brain and for developing hypotheses about cell- and circuit-level function.

[Editors' note: The authors have substantially addressed all the reviewers' concerns and comments.]

---

## [Author Response]

The following is the authors’ response to the previous reviews

**Recommendations for the authors:**

**Reviewer #1 (Recommendations for the authors):**
The authors have adequately addressed all of my concerns. I have no further questions or concerns.

We thank the Reviewer #1.

**Reviewer #2 (Recommendations for the authors):**
We thank the Reviewer #2 for thoughtful recommendations.(1) Figure 1A, 1B, 2B, 2C, etc.: The Y-axis label is confusing. I assume the intention was to make big numbers small by dividing by 1000. The comma makes the label confusing. Perhaps, make the label more "mathematical" as in "Avp density ((transcript/µm2) * 10-3)" or rearrange the math to be clearer as in "Avp density (transcript/1000 per µm2)".

Great suggestion and done exactly as suggested in Figures 1, 2 and 4.

(2) Figure 1B and 1C: The figure and legend do not match up. Either switch the figures or the legends. Currently, legend 1B describes image 1C.

Agreed and done as suggested.

(3) Figure 2A is broken up into separate pages/panels. It could be integrated better or separated to make A and B, then shift B and C to C and D.

Great suggestion and we have done exactly as suggested.

(4) Figure 2 legend: I recommend putting the scale bar info with (A) rather than at the end. The stars used in the figure are not explained in the legend.

Good points. We have made all necessary changes as suggested.

(5) Supplementary Figure 1B: The legend states that the data are the number of transcript-containing cells, but the figure states transcript number.

We thank the Reviewer for pointing out this typo. We corrected all graph legends in the Supplementary Figure 1.